# Pathophysiological Aspects of Muscle Atrophy and Osteopenia Induced by Chronic Constriction Injury (CCI) of the Sciatic Nerve in Rats

**DOI:** 10.3390/ijms24043765

**Published:** 2023-02-13

**Authors:** Francesca Bosco, Lorenza Guarnieri, Saverio Nucera, Miriam Scicchitano, Stefano Ruga, Antonio Cardamone, Samantha Maurotti, Cristina Russo, Anna Rita Coppoletta, Roberta Macrì, Irene Bava, Federica Scarano, Fabio Castagna, Maria Serra, Rosamaria Caminiti, Jessica Maiuolo, Francesca Oppedisano, Sara Ilari, Filomena Lauro, Luigi Giancotti, Carolina Muscoli, Cristina Carresi, Ernesto Palma, Micaela Gliozzi, Vincenzo Musolino, Vincenzo Mollace

**Affiliations:** 1Department of Health Sciences, Institute of Research for Food Safety and Health (IRC-FSH), University Magna Graecia of Catanzaro, 88100 Catanzaro, Italy; 2Department of Medical and Surgical Science, University Magna Grecia, 88100 Catanzaro, Italy; 3Laboratory of Pharmaceutical Biology, Department of Health Sciences, Institute of Research for Food Safety and Health (IRC-FSH) Center, University Magna Graecia of Catanzaro, 88100 Catanzaro, Italy; 4Henry and Amelia Nasrallah Center for Neuroscience, Department of Pharmacology and Physiology, Saint Louis University School of Medicine, Grand Blvd, St. Louis, MO 63104, USA

**Keywords:** chronic constriction injury, muscle atrophy, muscle atrophy model, osteoporosis, osteopenia, osteosarcopenia

## Abstract

Skeletal muscle atrophy is a condition characterized by a loss of muscle mass and muscle strength caused by an imbalance between protein synthesis and protein degradation. Muscle atrophy is often associated with a loss of bone mass manifesting as osteoporosis. The aim of this study was to evaluate if chronic constriction injury (CCI) of the sciatic nerve in rats can be a valid model to study muscle atrophy and consequent osteoporosis. Body weight and body composition were assessed weekly. Magnetic resonance imaging (MRI) was performed on day zero before ligation and day 28 before sacrifice. Catabolic markers were assessed via Western blot and Quantitative Real-time PCR. After the sacrifice, a morphological analysis of the gastrocnemius muscle and Micro-Computed Tomography (Micro-CT) on the tibia bone were performed. Rats that underwent CCI had a lower body weight increase on day 28 compared to the naive group of rats (*p* < 0.001). Increases in lean body mass and fat mass were also significantly lower in the CCI group (*p* < 0.001). The weight of skeletal muscles was found to be significantly lower in the ipsilateral hindlimb compared to that of contralateral muscles; furthermore, the cross-sectional area of muscle fibers decreased significantly in the ipsilateral gastrocnemius. The CCI of the sciatic nerve induced a statistically significant increase in autophagic and UPS (Ubiquitin Proteasome System) markers and a statistically significant increase in Pax-7 (Paired Box-7) expression. Micro-CT showed a statistically significant decrease in the bone parameters of the ipsilateral tibial bone. Chronic nerve constriction appeared to be a valid model for inducing the condition of muscle atrophy, also causing changes in bone microstructure and leading to osteoporosis. Therefore, sciatic nerve constriction could be a valid approach to study muscle–bone crosstalk and to identify new strategies to prevent osteosarcopenia.

## 1. Introduction

Skeletal muscle is one of the largest tissues in the human body, and it is fundamental to the generation of force and locomotion. The high plasticity of muscle tissue is due to its ability to adapt, both structurally and functionally, to external stimuli responsible for changes in muscle mass [1]. A lack of muscle mass and strength is associated with reduced quality of life, independence and life expectancy. Muscle loss is a condition associated with many diseases, such as heart failure, cancer, chronic obstructive pulmonary disease (COPD), diabetes and AIDS [2,3,4] and with many other chronic diseases, including chronic inflammation of the liver or pancreas, immobilization, prolonged bed rest, etc. [5].

Furthermore, muscle atrophy is often associated with osteoporosis, a bone tissue disorder characterized by reductions in bone density and strength. The co-existence of osteoporosis and muscle loss leads to the definition of “osteosarcopenia syndrome” with an increased risk of bone fracture, consequent hospitalization, immobilization and a reduction in quality of life [6,7]. Because there are no specific drugs for the treatment of this syndrome, the only therapeutic approach is mainly based on dietary supplementation accompanied by physical activity to increase the trophic stimulus on the musculoskeletal system [8,9]. However, in many situations, patients (such as those suffering a critical illness, fractures or nerve damage) are unable to perform physical activity, and immobility makes this remedy ineffective or otherwise inapplicable [10]. Although peripheral nerve damage is a clinical condition that might be caused by disease [11], physical trauma is the most common cause of injury to the nerves [12,13]. Animal models, with induced chronic constriction injury to the sciatic nerve [14] and its accompanying symptoms, such as neuropathic pain, motor dysfunction and skeletal muscle atrophy [14,15,16,17], are widely used to study peripheral nerve damage. Although CCI is a well-investigated model for the research of neuropathic pain in animals [14,15,16,17,18], very little is known about the musculoskeletal changes associated with this injury. Healthy muscle tissue is guaranteed by the balance between protein synthesis and degradation [19,20,21]. Generally, a loss of muscle mass can be attributed to a decreased rate of protein synthesis, an increased rate of protein degradation or both [22]. Increased protein degradation in atrophic muscle appears to be linked to the increased activity of two major protein degradation pathways: the autophagic pathway and the Ubiquitin Proteasome System (UPS) [17,18,19,20,21,22,23,24]. These pathways involve several atrogenes that can activate the muscle wasting program in many conditions, such as starvation, cancer cachexia, denervation, heart failure and aging [25,26].

However, in CCI-induced muscle atrophy, the catabolic machinery is activated, and protein synthesis rates are increased during nerve damage-induced muscle loss, suggesting a compensatory mechanism of muscle remodeling [15,16]. Contextually, in this scenario, regeneration ability mediated by muscle-specific stem cells is not guaranteed, and satellite cells remain in the quiescent phase without the possibility of differentiating into myoblasts [27,28,29]. Furthermore, bone loss has been recognized in connection with muscle disuse and dysfunction [30], but this can occur also in cases of nerve damage induced by crushing [31] or loose constriction of the sciatic nerve [32], which can lead to a change in bone mineral content (BMC) and bone mineral density (BMD) [32,33]. Although nerve injury may cause a direct alteration in bone–nerve interactions and a loss of BMD, it is not easy to isolate the direct effects of nerve injury on BMD, as constriction of the sciatic nerve causes immobilization, paralysis and a reduction in mechanical loading, which, in turn, reduces BMD.

In the present rat-based model study, we hypothesized that CCI of the sciatic nerve causes an effect on body weight and body composition over 28 days and exerts a modulation of the catabolic pathways in the GC muscle. Furthermore, we also hypothesized a modification of the microstructure of the tibial bone following damage to the sciatic nerve.

## 2. Results

### 2.1. Body Weight and Body Composition of Rats That Underwent Chronic Constriction Injury

We measured the body weight and body composition of animals to evaluate if CCI causes a change. Rats that underwent CCI had a lower body weight increase on day 28 compared to the naive group of rats (*p* < 0.001; Figure 1C).

Increases in lean body mass and fat mass were significantly lower in the CCI group compared with those of the non-operated control rats (*p* < 0.001; Figure 1A,B).

### 2.2. Biochemical Analysis after Chronic Constriction Injury

We evaluated the serum levels of some isoenzymes that can be released into the circulatory stream following muscle injury. Rats that underwent CCI had a higher serum level of Creatine Kinase (CK), Glutamic Oxaloacetic Transaminase (GOT) and Glutamic Pyruvic Transaminase (GPT) and lower levels of serum calcium (Ca^2+^) compared to naïve rats (Figure 2).

### 2.3. Constriction Injury of the Sciatic Nerve Induced Muscle Wasting in Rats

The CCI of the sciatic nerve had strong effects on individual muscles. The weights of the soleus muscle, extensor digitalis longus muscle (EDL) and tibialis muscle were significantly lower in the ipsilateral hindlimb compared with those of contralateral muscles in CCI animals (Figure 3).

Moreover, ipsilateral GC muscle weight was lower compared to that of contralateral muscle 28 days after CCI (Figure 4E). Sciatic nerve damage led to a fully developed muscle-wasting phenotype at 28 days post surgery (Figure 4A,B). Interestingly, the cross-sectional area (CSA) (Figure 4F) of muscle fibers visualized by hematoxylin and eosin (H&E) staining (Figure 4C,D) decreased significantly in the ipsilateral GC compared to that of the contralateral side in CCI rats.

Magnetic resonance imaging (MRI) images also showed a reduction in the volume of damaged hindlimb (Figure 5) and a reduction in individual ipsilateral muscles compared to those of the contralateral on day 28 (Figure 6).

### 2.4. E3 Ubiquitin Ligase Gene Expression Was Up-Regulated in Gastrocnemius Muscle after CCI of the Sciatic Nerve

To assess UPS hyperactivation, we examined the gene expression levels of *F-box32* (*Fbx32*) and *Tumor-Necrosis-Factor-Receptor-Associated Factor 6* (*TRAF-6*). As shown in Figure 7 mRNA expression levels of *Fbx32* and *TRAF-6* were higher in ipsilateral GC compared to those of contralateral muscle.

### 2.5. Catabolic Pathways in Rat Gastrocnemius Muscle after Chronic Sciatic Nerve Constriction Injury

The analysis of the catabolic pathways involved was focused on Beclin-1, microtubule-associated protein 1, light chain 3 beta isoform I (LC3B-I) and P62 to assess the involvement of the autophagy system, and on TRAF-6 and Fbx32 to assess the involvement of UPS. The evaluation of pax-7 expression levels was performed to assess muscle regeneration. The CCI of the sciatic nerve induced a statistically significant increase in Beclin-1 expression (Figure 8A).

A significant increase in LC3B-I was detected (Figure 8B). Moreover, the levels of p62, a marker of substrate sequestration into an autophagosome, were assayed. Levels of p62 were significantly higher in the injured ipsilateral skeletal muscle GC than those of the contralateral side (Figure 8C). In addition, levels of the protein TRAF-6 were higher in the ipsilateral GC than those of contralateral muscle (Figure 8D). Interestingly, the ubiquitin protein ligase Fbx32 was significantly up-regulated (Figure 8E). Western blot analysis showed that CCI of the sciatic nerve also induced a statistically significant increase in Pax-7 (Paired Box-7) expression in the ipsilateral GC of rats 28 days after surgery compared to that of the contralateral GC muscle (Figure 8F).

### 2.6. Increased Stem Cell Presence in Ipsilateral Gastrocnemius Evaluated with Pax-7 Immunostaining

Pax-7 immunostaining was performed to evaluate the contribution of muscle stem cells to CCI-induced atrophy. The immunohistochemical analysis (IHC) of ipsilateral gastrocnemius muscle revealed a higher number of Pax-7-positive satellite cells compared to that in contralateral muscle (Figure 9).

### 2.7. Increased Myogenic Markers Gene Expression in Gastrocnemius Muscle after CCI of Sciatic Nerve

Myogenic regulatory factors were assessed to evaluate the alteration of myogenic differentiation. mRNA expression levels of *myogenin* (Myog) and myosin were higher after CCI (Figure 10).

### 2.8. Bone Microstructure Observation of Tibia Bone following Constriction Injury of Sciatic Nerve

A Micro-CT analysis was performed to evaluate the architecture and bone structural parameters of the tibia bone. Micro-CT (Figure 11 and Figure 12 showed that many parameters, such as the BMD, bone volume fraction (BV/TV), bone surface density (BS/TV), trabecular number (Tb.N), connectivity density and bone perimeter (B.Pm) of the tibial bone on the ipsilateral hindlimb side were significantly lower than those of the contralateral tibial bone (Figure 13), whereas the structure model index (SMI) was significantly higher 28 days after the surgery (Figure 13). Moreover, calcium content showed a trend toward a reduction in the tibial bone on the ipsilateral hindlimb when compared to that of the contralateral side (*p* = 0.0879; Figure 13). This alteration in bone architecture parameters corresponds to a profile of osteoporosis [7].

### 2.9. Gene Expression Analysis in Femurs after CCI of Sciatic Nerve

We also evaluated the gene expression of some bone catabolic markers. The gene expression levels of *TRAF-6* were higher in ipsilateral bone compared to those of the contralateral; instead, Alkaline Phosphatase (*ALP*), osteoprotegerin (*OPG*), Receptor Activator of Nuclear factor Kappa beta Ligand (*RANKL*) and Runt-related transcription factor 2 (*RUNX2*) were lower in the ipsilateral femur (Figure 14).

### 2.10. Serum Markers Procollagen Type 1 Amino-Terminal Propeptide (P1NP) and (C-Terminal Collagen Cross-Linkage 1) CTX-1 Analysis after CCI of the Sciatic Nerve

To assess whether there was an alteration in bone turnover, we examined the serum levels of P1NP and CTX-1 markers. As shown by the ELISA analysis, in CCI rats, a significative reduction in CTX-1 serum levels and a trend toward significance for the P1NP serum marker compared to those of naïve animals were observed (Figure 15).

## 3. Discussion

Here, we demonstrated that (i) CCI injury of the sciatic nerve, performed in young rats, caused a change in overall weight and body composition. (ii) This change was associated with the skeletal muscle atrophy of the ipsilateral hindlimb driven by the hyperactivation of autophagy and UPS. (iii) The levels of Pax-7 were sustained 28 days after chronic constriction injury, suggesting that the differentiation of muscle cells could be impaired by CCI-induced atrophy and wasting. (iv) CCI injury also induced changes in bone microstructure, leading to osteoporosis.

CCI is a well-known model used to study peripheral nerve damage [14,15,16,17,18,19,20,21,22,23,24,25,26,27,28,29,30,31,32,33,34,35], which is accompanied by debilitating symptoms, such as neuropathic pain and hampered motor function. However, at present, very little is known about musculoskeletal and osteopathic changes associated with this injury, even though it is a widely used model for neuropathic pain research [36].

It is known that skeletal muscle contains isozymes of CK, GOT and GPT, which can be released into the bloodstream following muscle injury [37,38]. Pathological conditions that cause damage to muscle energy production can also increase blood levels of CK, GOT and GPT, reflecting damage to myofibers. CK is an enzyme expressed in muscle tissue that, using high amounts of adenosine triphosphate (ATP), catalyzes the reversible conversion of creatine to create phosphocreatine (PCr) and adenosine diphosphate (ADP). In this circuit, cytosol isoenzymes are closely coupled to glycolysis and produce ATP for muscle activity. Following muscle and sarcolemma damage, there is leakage of these enzymes from the cytoplasm, resulting in increased serum levels [39]. In agreement with the studies reported in the literature, our results show increased levels of CK, GOT and GPT in the CCI group as a consequence of sciatic nerve ligation, which led to the loss of muscle strength and altered muscle metabolism, and most importantly, high levels of these markers reflected increased protein catabolism, which was accompanied by the loss of body weight and damage to myofibers.

In this experiment, the group that underwent ligation of the sciatic nerve had a significantly lower body weight on day 28 of the experiment compared to that of the control group, which gained less weight than expected, according to guidelines provided by the breeder (Envigo Harlan).

Our results are consistent with previous studies that reported that body weight in rats subjected to CCI surgery is less compared to sham groups over a 14-day period [40]. In our study, rats belonging to the CCI group showed an atrophic phenotype of GC ipsilateral muscle, as shown by representative photos [36,37,38,39,40,41]. Atrophy induced by ligation of the sciatic nerve is well documented. Indeed, other groups have reported atrophy of the soleus [15,16,17,18,19,20,21,22,23,24,25,26,27,28,29,30,31,32,33,34,35,36,37,38,39,40,41,42], tibialis, EDL [15], GC and rectus femoris muscles [43] after peripheral nerve injury.

Moreover, the total number of fibers indicates a reduction in the CSA area in ipsilateral muscle and subsequent muscular atrophy due to the loss of mass in individual fibers, which could be consistent with muscular atrophy rather than dystrophy [44]. Furthermore, the MRI image also indicates a reduction in mass of the damaged hindlimb, and manual muscle segmentation showed a reduction in individual ipsilateral muscle volume compared to that of the contralateral.

Sciatic nerve ligation is a loose ligation that causes epineural swelling and inflammation, but it leaves the axons intact [14,15,16,17,18,19,20,21,22,23,24,25,26,27,28,29,30,31,32,33,34,35,36,37,38,39,40,41,42,43,44,45,46]; therefore, the amount of muscle denervation occurring in CCI is less than that in other more severe injury models, such as crush injury [47]. The reduced activity in CCI rats post surgery suggests that hypokinesia plays a very important role in muscle atrophy. Earlier studies have shown that muscle atrophy occurs in soleus, EDL, GC and tibialis muscles with cast immobilization in mice [48], as well as with tail cast suspension [49], space flight and whole-body suspension in rats [50] and in sciatic injured models [51]. Generally, a loss of muscle mass can be due to a decreased rate of protein synthesis, an increased rate of protein degradation or both. There is a consensus that, in many chronic diseases, such as cachexia, COPD or heart failure, atrophy is caused by sustained proteolysis. Although in disuse atrophy and immobilization in humans, a decrease in myofibrillar protein synthesis appears to be the predominant mechanism believed to cause muscle loss [51,52]. In recent studies, the activation of proteolytic systems in muscle immobilization has also become of great interest [53,54,55]. The up-regulation of autophagy and the UPS has been implicated in several models of heart and skeletal muscle wasting [56]. Later work has shown sustained autophagy as well as activation of the UPS after nerve transection [57]. However, although a lot of work has been conducted on nerve transection, less is known about CCI to the sciatic nerve. Here, we report that the hyperactivation of autophagy and UPS occurred after CCI. We found that, in the skeletal muscle of the damaged hindlimbs, autophagy was associated with increases in Beclin-1, LC3B-I and p62 accumulation. Protein p62 has a principal role in the recognition of ubiquitinylated proteins or depolarized mitochondria during selective autophagy; interestingly, it was described that p62 can also deliver ubiquitinylated cargos to the proteasome [58].

Regarding proteins that play a regulatory role in the activation of signaling cascades related to muscle atrophy, TRAF-6 might potentially be an upstream regulator for the activation of pathways involved in the loss of muscle proteins in conditions of wasting [17]. TRAF-6 expression is increased in several models of muscle atrophy, including fasting and cancer, leading to the downstream activation of major catabolic pathways in skeletal muscle, including autophagy [17]. In accordance with these reports, in the GC CCI model, we found an increase in *TRAF-6* gene expression and increased high levels of TRAF-6 protein. Moreover, the protein content of E3 ubiquitin ligase, Fbx32, up-regulated in the denervated muscle [59], was also up-regulated in the CCI model in our study, underlying common pathways of protein degradation in different models of muscle wasting [60]. The gene expression of *Fbx32* evaluated with PCR analysis was found to be up-regulated in the GC muscles of the muscle atrophy models induced by both starvation and denervation [61]. This was also confirmed in our model of atrophy induced by ligation of the sciatic nerve. Moreover, the protein content of E3 ubiquitin ligase, Fbx32, up-regulated in the denervated muscle [59], was also up-regulated in the CCI model in our study, underlying common pathways of protein degradation in different models of muscle wasting [60].

Skeletal muscle regeneration is attributed to satellite cells, muscle stem cells located beneath the basal lamina surrounding each myofiber. The transcription factors of the *Paired box (Pax)* gene family, Pax-3 and Pax-7, are critical for the potential self-renewal of satellite cells [62]. Pax-7 protein is known to be expressed in quiescent satellite cells, maintained during progression from quiescent to activation and proliferation and then lost during differentiation. Pax-7 induces the expression of genes responsible for promoting proliferation in the myogenic process and represses genes that drive differentiation. We report that, in the GC of the injured hind limb, Pax-7 levels were sustained 28 days after chronic constriction injury. Furthermore, the immunohistochemical analysis of Pax-7-positive satellite cells showed that the number of satellite cells per mm^2^ was greater in the ipsilateral muscle than that in the contralateral gastrocnemius muscle [63]. The satellite cell activation process, in which resident stem cells migrate to the injury site and begin to proliferate, partially losing their myoblastic characteristics, can be assessed with several myogenic regulatory factors. These factors include the Myogenic determination factor (MyoD) and Myogenin, which are downstream from Pax-7 and promote myogenic differentiation [64]. Myogenin, a direct target of MyoD, initiates the terminal differentiation of myogenic progenitor cells, which is accompanied by the down-regulation of MyoD expression [65]. *Myog* gene expression levels, assessed as a terminal marker of differentiation by qPCR, were found to be up-regulated by qPCR analysis in this experimental model. Damage to the environment surrounding the satellite cell causes the deterioration of the basal lamina and its exit from the state of quiescence, causing its activation [66]. Normally, quiescent satellite cells are characterized by their expression of Pax-7 but not of myogenin [67]. In the experimental model of peripheral nerve damage, myogenin is up-regulated in skeletal muscle and regulates the expression of *Fbx32*, which promotes proteolysis and atrophy. It is known that, regardless of the muscular activity carried out, nerve–muscle contact is sufficient to suppress the proliferation of satellite cells, and it seems that this effect can be mediated by myogenin itself [64]. Fbx32 is a target of Myog; the activation of the *Fbx32* promoter in tissues and cells is Myog-dependent and dependent on denervation [68]. Here, we observed that sciatic nerve ligation caused an increase in *myogenin* gene expression in the gastrocnemius muscle, and this finding is consistent with the increase in *Fbx32* observed by us, both as gene expression and as protein expression. Initially, the proliferating myoblasts differentiate into elongated myocytes, which fuse to form nascent myotubes and then further fuse with additional myocytes or other myotubes to generate mature myotubes that align with each other, constituting the myofibers, the contractile units of skeletal muscle [69]. Newly formed myofibers are characterized by centrally located nuclei and by the expression of devMHC (developmental myosin heavy chain), a myosin heavy chain that is otherwise only expressed during embryonic development. Once the differentiation of myoblasts into myocytes occurs, they fuse to form syncytial myotubes which begin to express genes encoding sarcomeric proteins, such as MHC (myosin heavy chain). Conditions of muscular disuse lead to tissue atrophy characterized by alterations and in the passage of muscle fibers from type I to type II [70]. Indeed, muscle damage from denervation, hind limb suspension and spaceflight is known to cause a significant reduction in MHC I (myosin heavy chain type I) and an increase in MHC II (myosin heavy chain type II), consistent with the phenotypic transformation of slow fibers into fast fibers. In the chronic sciatic nerve constriction model, we also observed increased gene expression of MHC II. This increase, together with the increase in other differentiation markers, indicates an alteration in the process of muscle regeneration following an injury and a phenotype of atrophy [71].

Our findings support that differentiation is impaired in CCI and that such impairment can be a contributor in wasting [72].

Bone mass and skeletal muscle mass are controlled by several factors, such as genetics, diet, growth factors and mechanical stimuli. Increased mechanical loading of the musculoskeletal system stimulates an increase in the mass and strength of skeletal muscle and bone, whereas reduced mechanical loading and disuse rapidly promote a decrease in musculoskeletal mass and strength, which could lead to muscle atrophy and osteoporosis [73].

Bones are constantly being remodeled. Calcium moves in and out of them, and its balance generally reflects the degree of coupling of bone formation and resorption processes. It is important to recognize that, because 99% of body calcium is found in bone, it is not possible to build bone without a positive calcium balance, nor is there a negative balance without losing bone [74,75]. Calcium is involved in many biological processes, and its deficiency can cause osteoporosis [76,77]. Low calcium intake, inactivity and decreased muscle and bone mass have a negative impact on muscle and bone health [78].

When serum-free (ionized) calcium levels are low, the parotid gland is automatically stimulated, and Parathyroid hormone (PTH) is released to obtain calcium release from the bones and maintain the blood calcium level. If this happens repeatedly, eventually, the bones become porous and brittle, and more importantly, they become more susceptible to breakage [79]. In this study, we observed a significant decrease in serum calcium levels, reflecting a negative balance in the CCI group as a consequence of muscle atrophy, leading to osteopenia.

Weight-bearing bones are particularly sensitive to the absence of mechanical loads, with the proximal femur and tibia exhibiting ∼5% and 23% reductions in bone mineral density following 3 and 6 months of disuse, respectively [80]. Here, via analysis of micro-CT, we showed that osteoporosis occurred in the ipsilateral tibial bone in rats with sciatic nerve ligation. An experimental study on mononeuropathy in rats described changes in BMC and BMD in the ipsilateral tibial bone [81]. We extended this previous report by analyzing new parameters. In our study, at 4 weeks post surgery, many parameters indicating bone frailty, such as bone and calcium content, showed a trend toward a reduction. Furthermore, we analyzed some osteoporosis-related genes, such as *ALP*, *OPG*, *RUNX2*, *RANKL* and *TRAF-6*. However, in the ipsilateral femurs of animals subjected to CCI, we observed only a significant increase in *TRAF-6* gene expression, which could cause an alteration in physiological bone remodeling [82]. In fact, the deletion of *TRAF-6* is associated with an increase in the number of osteoblasts and a reduction in osteoclasts, as well as an increase in bone mineral density [83]. Two possible mechanisms may be involved in the pathogenesis of osteoporosis/osteopenia associated with CCI.

One is the partial immobilization of the hindlimb. Previous reports have shown that osteopenia develops within 10–21 days in mice with cast immobilization, accompanied by diminished BMC and decreases in dry bone mass, bone mineral density and metaphyseal bone volume [84]. The osteopathy seen in mice with cast immobilization is quite similar to the osteopathy that we report in this study. In addition to immobilization, nerve injury could also be directly responsible for osteopathy. Bone is densely innervated to the periosteum of the metaphyseal region, and the primary afferent axons that innervate the bone contain neuropeptides [85]. If such neuropeptides are released from the nerve and affect bone metabolism, osteoporosis could be related to their activity. Loose ligation of the sciatic nerve in rats induced an inflammatory response in the ipsilateral hind paw; hence, an increase in myeloperoxidase activity in the muscle, as well as increased skin blood flow and edema in the ipsilateral hind paw on day 4 after surgery, have already been described. Neuropeptides such as substance P, released from nerve terminals in injured axons, cause the increased synthesis and release of monocyte-derived cytokines [86]. These cytokines can also induce osteoclastogenesis [87] and, therefore, may subsequently cause osteoporosis.

Because the mRNA levels of the analyzed genes were not very informative, to analyze whether the observed osteopenia phenotype was due to catabolic bone loss or reduced bone formation, we measured the level of serum markers of these processes. The International Osteoporosis Foundation (IOF) recommends that a marker of bone formation and a marker of bone resorption be used as a reference for the assessment of bone turnover [88]. Bone turnover markers such as serum P1NP and CTX-1 can be useful in predicting future fracture risk [89]. Type I collagen is an important component of the bone matrix, and its precursor, type I procollagen, is secreted by osteoblasts during bone formation. Extension peptides at the ends of the procollagen molecule, P1NP and type I C procollagen pro-peptide (P1CP), are cleaved by enzymes during bone matrix formation and are released into the circulation [84]. In the present study, serum levels of P1NP decreased after CCI, and these data indicate that osteogenesis is disfavored in this experimental model. Contextually, serum levels of CTX-1 were low; therefore, it would appear that the osteopenic phenotype was due more to decreased bone anabolism than to increased catabolism at 28 days after injury. It is known that, to occur, the process of bone resorption requires adequate numbers of osteoclasts accessing bone minerals. However, these osteoclasts must be activated by a mechanism that depends on previous osteoblastic stimulation [90]. Therefore, the fact that osteoclast activity is not found to be increased, as is the case in some experimental models of osteopenia, due, for example, to aging [91], could also be closely related to the experimental phase in which the analysis is performed. Further studies would be needed to observe the trend of the expression of this marker at later times. However, also in a study on osteoporosis carried out comparing smoking and non-smoking patients, it was found that smokers are more predisposed to osteopenia and show a lower level of the bone formation marker P1NP but also a more attenuated level of bone resorption marker CTX-1, and in this study, lower bone turnover was associated with lower BMD, as also happened in our experimental model [92].

However, it is known that peripheral nerve injury alters bone homeostasis and accelerates osteoporosis also through altered microvessel distribution. There is evidence that the bone vasculature is a crucial partner in the bone remodeling process and plays a role in resorption/formation coupling [93]. Alterations in intraosseous microvessel parameters, therefore, may be a contributing factor in the pathogenesis of post-injury osteoporosis [94].

For example, Vogt and Alagiakrishnan showed that bone blood inflow in people with osteoporosis or osteopenia is relatively lower than that in people with normal bone mass, indicating that bone blood inflow and bone mineral density are highly correlated [95,96]. In addition, impaired angiogenesis decreases trabecular bone formation and hypertrophic zone expansion in growth cartilage; consequently, the interruption of blood supply to the bone causes a marked decrease in bone density and strength [97].

Endothelial cells communicate with osteoprogenitor cells, both during development and during the healing process of bone fractures; thus, bone homeostasis depends on tissue vascularization. It is believed that osteoblasts can originate from circulating or perivascular osteoprogenitors, and that blood vessels carry osteoclast precursors to the surface of the bone where remodeling is about to take place [98].

Thus, endothelial dysfunction implicated in osteopenia due to nerve damage could, in our model of sciatic nerve ligation, affect the profile of osteopenia by causing a reduction in osteoblast levels but also by failure to increase osteoclasts.

Although further experiments are needed to better characterize the mechanisms underlying muscle atrophy and osteopathy, chronic sciatic nerve constriction appears to be a valid approach to study muscle–bone crosstalk and for the identification of new and more effective strategies to prevent osteosarcopenia.

The limitation of this model was the inability to rule out that there was a degree of variation among the rats subjected to CCI, due to variability in the tightness of the constrictions of tied knots with the sutures. However, to partially overcome this limitation, the intervention was performed again by the same researcher.

## 4. Materials and Methods

### 4.1. Animals

Adult male Wistar Han 8-week-old rats (*n* = 58) weighing 250 g ± 4 g were randomly divided into two groups: the naïve group (*n* = 10) and the CCI group (*n* = 48). On day 0, sciatic nerve ligation was performed on the right hindlimb of all rats belonging to the CCI group. The left (contralateral) hindlimb was operated on but without ligation of the sciatic nerve.

One day before surgery, baseline body weight and body composition were assessed. The same parameters were assessed once per week until the end of the study. MRI was performed on day 0 before ligation and day 28 before sacrifice. On the day of the sacrifice, animals were anesthetized and euthanized, and muscles were rapidly removed, weighed and immediately frozen in liquid nitrogen or fixed in 10% formalin for further analysis. After the sacrifice, Micro-computed tomography was performed on the tibiae.

All the experimental procedures conducted on animals were performed according to protocols approved by the Animal Care of University Magna Graecia of Catanzaro (957/2017-PR) and were carried out in compliance with the ARRIVE guidelines. All experiments were performed in accordance with the European Commission guidelines (Directive 2010/63/EU) for animals used for scientific purposes.

### 4.2. Chronic Constriction Injury

The surgery was performed under anesthesia, induced with 5% isoflurane in oxygen and then maintained with 2% isoflurane in oxygen. Anesthetized animals were monitored constantly during the surgery. On the right hindlimbs (ipsilateral), the sciatic nerve was exposed by making a skin incision and cutting through the connective tissue, 3–4 mm below the femur, between the gluteus superficialis and biceps femoral muscle. Four loosely constrictive ligatures of 4.0 chromic gut were tied around the right sciatic nerve with a double knot, 1 mm apart, proximal to the trifurcation of the sciatic nerve, to occlude but not arrest epineural blood flow. An identical surgery, without sciatic nerve damage, was performed in the left hindlimb (contralateral). Staples were used to suture (Autoclip, 9 mm), and the wounds were disinfected using an iodine solution (Riodine). After the surgery, the rats were housed individually in cages [14].

### 4.3. Analysis of Blood-Based Endpoints

A volume of around 1 mL of whole blood was collected into untreated Eppendorf tubes and centrifuged for 10 min at 2000 rcf at 4 °C. The serum was then collected and stored at −80 °C for later analysis. GOT, GPT, CK and Ca^2+^ were measured in serum samples using an automatic chemistry analyzer, XL-640 (Erba Mannheim), and the following Erba liquid stable reagents: GPT (ALT/GPT 330, XSYS0017), GOT (AST/GOT 330, XSYS0016), creatine kinase (CK 110, XSYS0022) and calcium (Ca^2+^ 100, BLT00015).

### 4.4. Body Composition Analysis

Total body fat, lean mass and body fluids were measured using a nuclear magnetic resonance spectroscopy device, EchoMRI-700TM (Echo Medical System, Houston, TX, USA), as previously described [18].

### 4.5. Magnetic Resonance Imaging

Before magnetic resonance imaging, rats were anesthetized with 4% isoflurane (Forane, Abbott), vaporized in oxygen (flow: 2 L/min) and then kept during the acquisition between 2 and 3% to maintain a breathing rhythm between 40 and 60 breaths per minute. All vital parameters were monitored through a multiparametric monitor. Body temperature was maintained at 37 °C. MRI images were acquired with a Bruker Pharmascan 70/16 US 7 Tesla bore MR scanner (Bruker Biospin MRI GmbH, Ettlingen, Germany), equipped with a total body transmitter–receiver coil. Axial T2_turboRARE weighted images with fat suppression were acquired with the following parameters: relaxation time (TR), 800.000 ms; echo time (TE), 25.00 ms; averages, 8; slices, 9; thickness of 1 mm; repetitions, 1; and field of view, 58 × 58 mm^2^.

### 4.6. Muscle Segmentation

Manual segmentation was performed using OsiriX imaging software (v. 12.0.2, Pixmeo SARL, Switzerland). The margins of individual muscles in Axial T2_turboRARE weighted images were manually traced through the “draw tool”, as defined in Greene’s Rat Anatomy Atlas [99] and in a more recent experiment performed by Zhang et al. [100]. The slice used was the same for all animals, and it represented the exact distance between the epiphysis and diaphysis. To improve the viewing of individual muscle margins, a Default WL/WW, “Vr Muscles-Bones Clut” and a Logarithmic opacity table were used.

### 4.7. Protein Extraction and Western Blot

An amount of 50 mg of GC muscle were homogenized in 500 μL of ice-cold lysis buffer (20 mM Tris–HCl, pH 7.5; 150 mM NaCl; 1 mM EDTA; 1 mM EGTA; 1% Triton X-100; 2.5 mM Na_4_P_2_O_7_; 20 mM NaF; 1 mM dithiothreitol; 1 mM Na_3_VO_4_; 1 mM β-glycerophosphate; and 10 μL/mL freshly added protease and phosphatase inhibitor cocktails), centrifuged at 14,000× *g* for 20 min at 4 °C, and supernatant was collected. A total of 20 μL of the supernatant was used to determine the total protein concentration with a Bradford assay (Bio-Rad, Hercules, CA, USA) using bovine serum albumin as a standard. Proteins were heat denatured for 5 min at 95 °C in sample loading buffer (500 mM Tris/HCl, pH 6.8; 30% Glycerol; 10% sodium dodecyl sulfate; 5% β-mercaptoethanol; and 0.024% bromophenol blue), and 30 μg of protein lysate was resolved by sodium dodecyl sulfate-polyacrylamide gel electrophoresis and transferred to nitrocellulose membranes (GE Healthcare, 10600001). After incubation, in a blocking solution (5% dry non-fat milk, Sigma-Aldrich, St. Louis, MO, USA), membranes were incubated overnight at 4 °C, shaking with the following primary antibodies: Beclin-1 (3738, Cell Signaling Technology, Boston, USA), LC3B-II/LC3B-I (PM-036, MBL, Woburn, MA, USA), p62 (M162-3, MBL, Woburn, MA, USA), TRAF-6 (Ab 33915, Abcam, Cambridge, UK), Fbx32 (ab168372, Abcam, Cambridge, UK), Pax-7 (MABD20 Merk Millipore, Darmstadt Germania) and GAPDH (ab181602, Abcam, Cambridge, UK). All antibodies were used at a concentration of 1:1000. Membranes were then washed in TBS (pH 7.6) with 0.1% Tween-20 and were incubated with a secondary antibody for 1 h at RT with shaking [23].

### 4.8. Histological Analysis

GC muscles of rats were harvested and fixed in 10% formalin for 24 h and were paraffin embedded for morphological analysis of the cross-sectional area of muscle fibers. For the assessment of tissue morphology, 5 μm thick sections of muscles were stained with H&E and visualized at room temperature (RT) on a microscope (Olympus bx53, U-LH100HG) using a digital camera (Nikon) at 20× magnification and Olympus cellSens Dimension 1.1 Software. The images, stored as JPEG files, were equally adjusted using Photoshop CS2 software (Adobe). For each muscle, the distribution of fiber CSA was calculated by analyzing 400 myofibers.

### 4.9. Immunohistochemistry

Formalin-fixed, paraffin-embedded sections (5 μm thick) were deparaffinized via exposure to BioClear and graded alcohols and were then washed in water. Epitopes were retrieved by heating the slides to 98 °C for 30 min in 10 mM Sodium Citrate buffer (pH 6.0). Sections were washed 2 times with Wash Buffer (Tris 25 mM, NaCl 0.15 M, and Tween-20 0.05%). Immunohistochemistry was performed using Mouse and Rabbit Specific HRP Plus Detection IHC Kit (ab93697). Slides were incubated with hydrogen peroxide (H_2_O_2_) for 30 min at RT, washed 3 times in Wash Buffer, subsequently blocked with Protein Block for 1 h at RT to block nonspecific background staining and were finally incubated with primary antibody anti-Pax-7 (NBP2-32894, 1:50) overnight at 4 °C in a humidified chamber. The next day, the slides were rinsed in Wash Buffer (2 × 3 min) and covered with secondary biotinylated goat anti-polyvalent plus sera (Mouse and Rabbit Specific HRP Plus Detection IHC Kit, Ab93697) for 20 min (RT). Then, the sections were washed 2 times (RT) in wash buffer and incubated with streptavidin peroxidase plus (Ab93697-Mouse and Rabbit Specific HRP detection IHC Kit; Abcam, USA) for 20 min (RT). 3,3ʹ-diaminobenzidine (DAB; Vector Laboratories, Newark, CA, USA) as a chromogen was applied for the visualization of primary antiserum. The working solution of DAB was added to the slides, and the process was observed under an LM (light microscope). Finally, the slides were rinsed in distilled water. Counterstaining was performed with Mayer’s hematoxylin. After washing in distilled water, the slides were dehydrated, mounted in Mounting Medium (ab 1859351, USA) and coverslipped [101].

### 4.10. Micro-Computed Tomography

After the sacrifice, the tibiae were explanted from the animals and stored in alcohol at a temperature of 4 °C. Right and left tibiae were scanned using a SkyScan 1176 (SkyScan, Kontich, Belgium). The explanted bones were removed from alcohol storage and dried superficially on paper tissue before being wrapped in plastic “cling-film” or in parafilm to prevent drying during scanning (and associated movement artifacts). Each plastic-wrapped bone was placed in a plastic/polystyrene foam tube that was mounted horizontally in the 1176 scanner sample chamber for micro-CT imaging. Reconstruction was carried out using the Skyscan Nrecon2 software, which facilitates network-distributed reconstruction carried out on four PCs running simultaneously. The time needed for the reconstruction of the dataset scan is usually much less than the scan duration.

Trabecular bone was assessed in 400 slices of proximal tibiae (immediately distal to the epiphyseal plate). The settings were a source voltage of 65 KVp and a source current of 380 μA. The trabecular bone was manually segmented. Parameters are reported according to published guidelines. Trabecular parameters included bone volume fraction (BV/TV), number (Tb.N), thickness (Tb.Th), separation (Tb.Sp), structure model index (SMI) and connectivity density (Conn.D) [102].

### 4.11. Bone Calcium Content

Right and left tibiae were isolated and cleaned of soft tissue. The bones were dried at 110 °C for 6 h and weighed and were then ashed at 800 °C for 4 h, weighed again, and dissolved in 1 mL of 6N HCl. Calcium content was determined via colorimetric determination with a Quantichrom calcium Assay kit (BioAssay Systems, Hayward, CA, USA).

### 4.12. Reverse Transcription and Quantitative Real-Time PCR (qRT-PCR) Performed on GC Muscle Tissue

Total RNA from GC muscle was extracted with Trizol reagent (Gibco, Life Technologies, Carlsbad, CA, USA) according to the manufacturer’s instructions. The RNA quantity and quality were assessed with the NanoDrop ND-2000 Spectrophotometer (Waltham, MA, USA). To evaluate transcript changes, 1000 ng of total RNA was reverse-transcribed to cDNA using the “High Capacity cDNA Reverse Transcription Kit” (Applied Biosystems, Carlsbad, CA, USA).

The following TaqMan gene expression assays (Applied Biosystems, Carlsbad, CA, USA) were used to detect and quantify genes using the Viia7 DX real-time PCR instrument (Life Technologies, Waltham, MA, USA): *TRAF-6* (Rn00590197_m1), *Fbx32* (Rn00591730_m1), *myosin* (Myh2) (Rn01470656_m1), *myogenin* (MyoG) (Rn00567418_m1) and *GAPDH* (Rn01462661_g1).

### 4.13. Reverse Transcription and Quantitative Real-Time PCR (qRT-PCR) Performed on Femur Bones

Rat femurs were homogenized using gentleMACS Dissociators (Miltenyi). Total RNA from femurs was extracted with Trizol reagent (Gibco, Life Technologies, Carlsbad, CA, USA) according to the manufacturer’s instructions.

RNA quantity and quality were assessed with the NanoDrop ND-2000 Spectrophotometer (Waltham, MA, USA). To evaluate transcript changes, 1000 ng of total RNA was reverse-transcribed to cDNA using the “High-Capacity cDNA Reverse Transcription Kit” (Applied Biosystems, Carlsbad, CA, USA). The mRNA expression of *TRAF-6*, *ALP*, *RANKL*, *OPG*, *RUNX2* and *β-ACTIN* was quantified via real time-PCR using SYBR^®^ Green dye (SYBR^®^ Green PCR Master Mix, Applied Biosystems, Foster City, CA, USA) (see Table 1).

### 4.14. Real-Time PCR Data Analysis

To carry out the analysis of data obtained from real-time PCR, the arithmetic mean of the Ct (Cycle Threshold) in triplicate values was determined. ΔCt was calculated as the difference between the Ct of the reference gene and the Ct of the target gene. ΔΔCt was calculated as the difference between the mean of the ipsilateral ΔCt and the mean of the contralateral ΔCt. The statistical significance was assessed based on the fold change values calculated as 2^ΔΔCt^ [103].

### 4.15. ELISA Analysis

Rat P1NP and CTX-I serum levels were detected with ELISA kits (P1NP MBS2506450 and CTX-1 MBS728629, San Diego, CA, USA). In ELISA microplate wells, serum samples were added and combined with the specific antibody. A biotinylated detection antibody specific to P1NP, CTX-I and Avidin-Horseradish Peroxidase (HRP) conjugate was added to each microplate well and incubated. Free components were washed away, and the substrate solution was added to each well. The wells that contained P1NP, CTX-I, biotinylated detection antibody and Avidin-HRP conjugate appeared blue in color. The reaction enzyme-substrate was stopped by the addition of Stop Solution, and the color turned yellow. Optical density (OD) was measured spectrophotometrically at a wavelength of 450 nm. In the samples, the concentrations of P1NP and CTX-I were calculated by comparing the OD of the samples with the standard curve using the specific software for ELISA Data analysis “Curve Expert 1.4”.

### 4.16. Statistical Analysis

The data were analyzed with GraphPad PRISM 9.1.2 (Graph Pad Software, Inc., La Jolla, CA, USA). All data are expressed as mean ± S.E.M. Normally distributed data of body weight and body composition were analyzed via a one-way ANOVA, followed by Tukey’s test, whereas data not normally distributed were analyzed using a Kruskal–Wallis analysis of variance, followed by Dunn’s tests or a Dunn’s test. A paired Student’s *t*-test was used for statistical muscle weight data analysis, cross-sectional area of fibers, Western blot analysis and bone parameters. PCR data were analyzed by fold change and tested with a paired Student’s *t*-test. Values with *p <* 0.05 were considered statistically significant. Values with *p <* 0.09 were considered statistically tending toward significance.

## Figures and Tables

**Figure 1 ijms-24-03765-f001:**
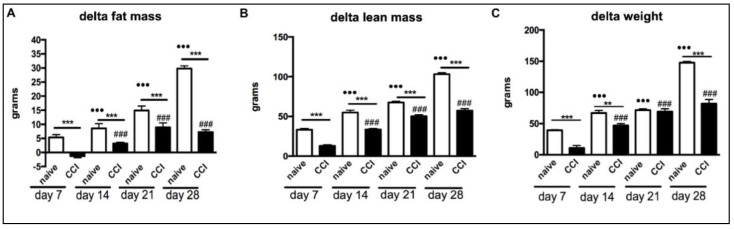
Effects of CCI of the sciatic nerve on weight and body composition. CCI animals showed a reduced increase in body weight, fat mass and lean mass compared to naïve rats. Variations in fat mass (**A**), lean mass (**B**) and body weight (**C**) are presented as the absolute difference between baseline values and the values observed every seven days after the intervention. The results are expressed as mean ± S.E.M. •••: *p* < 0.001 vs. Naive day 7; ###: *p* < 0.001 vs. CCI day 7; **: *p* < 0.01 at 14 days; ***: *p* < 0.001 at 7 and 28 days.

**Figure 2 ijms-24-03765-f002:**
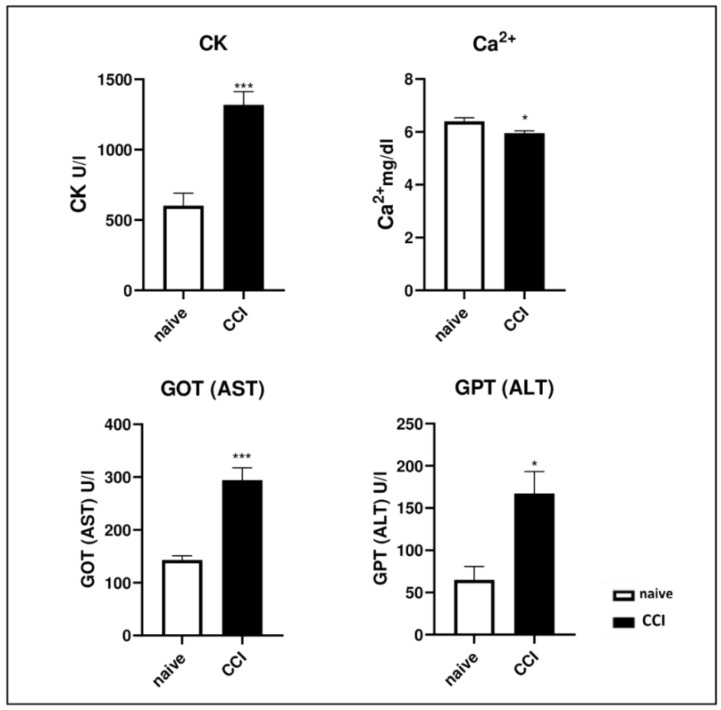
Effects of CCI of the sciatic nerve on serum biochemical marker levels. CK, GOT and GPT levels were higher in CCI rat serum compared to those of naïve rats. Calcium serum levels, instead, were reduced in rats that underwent chronic constriction injury. The results are expressed as mean ± S.E.M. *: *p* < 0.05, ***: *p* < 0.001 vs. naϊve.

**Figure 3 ijms-24-03765-f003:**
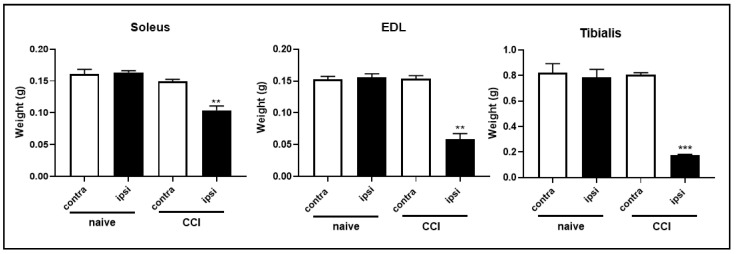
Skeletal muscle weight at the end of the study. Ipsilateral skeletal muscle weight in animals subjected to intervention was always lower compared to that of contralateral muscles. The results are expressed as mean ± S.E.M. **: *p* < 0.01, ***: *p* < 0.001 vs. contralateral.

**Figure 4 ijms-24-03765-f004:**
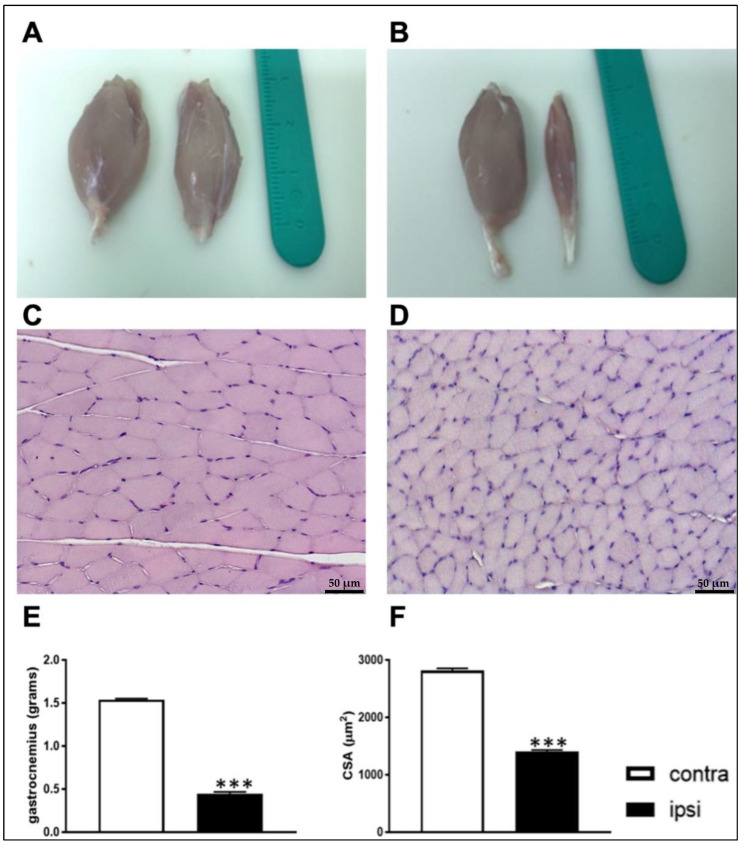
GC muscles of rats, subjected to CCI, exhibited an atrophic phenotype on day 28. (**Top**) Representative photos of GC muscles of a naïve animal (**A**) and a CCI animal (**B**) explanted on day 28 of the experiment showed a muscle-wasting phenotype. (Middle) Representative haematoxylin and eosin (H&E) staining images of contralateral GC (**C**) and ipsilateral gastrocnemius (**D**) showed a reduction in the CSA of muscle fibers in CCI. Scale bar: 50 µm. (**Bottom**) GC weight on day 28 was reduced in ispilateral hindlimb compared to that of the contralateral (**E**); also, CSA of muscle fibers was reduced in ipsilateral GC compared to that of the contralateral (**F**). The results are expressed as mean ± S.E.M. ***: *p* < 0.001 vs. contralateral.

**Figure 5 ijms-24-03765-f005:**
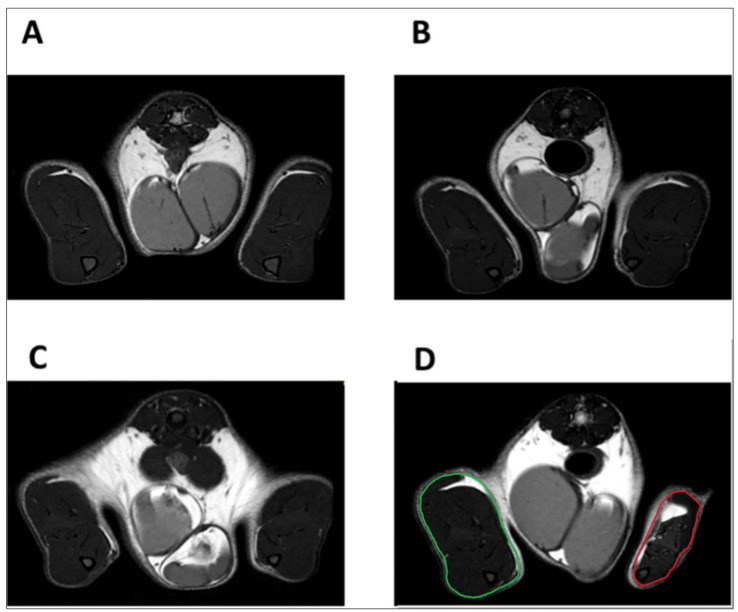
Representative axial T2_turboRARE weighted images with fat suppression of rat hindlimbs. Naïve rat on day 0 (**A**) and day 28 (**B**); CCI rat on day 0 (**C**) and day 28 (**D**). In Figure 5D, the red line delineates the area corresponding to the entire right paw (ipsilateral) of the CCI animal after 28 days, which was found to have a reduced volume. In contrast, the perimeter of the contralateral paw (normal volume) is delineated in green.

**Figure 6 ijms-24-03765-f006:**
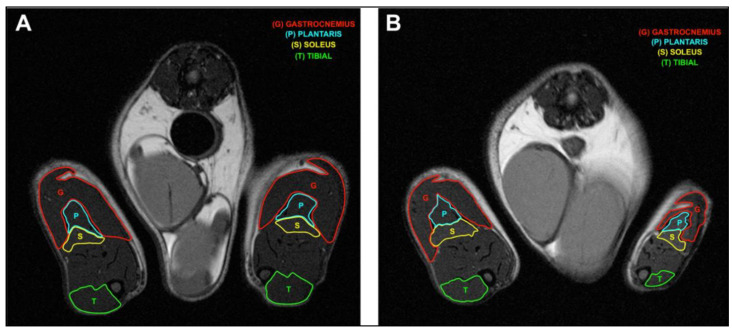
Manual segmentations of individual muscles in axial T2_weighted images with fat suppression of rat hindlimbs. Naïve rat (**A**) and CCI rat (**B**) on day 28.

**Figure 7 ijms-24-03765-f007:**
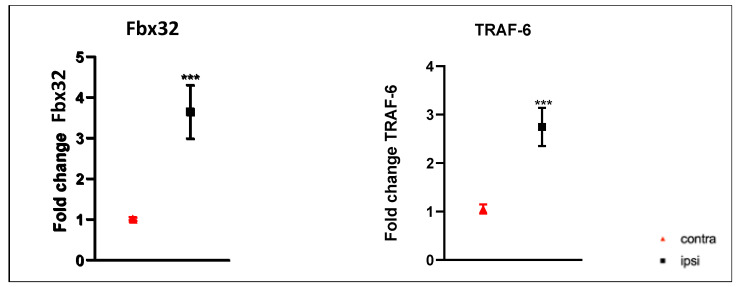
Results of real-time PCR. Fbx32 and TRAF-6 gene expression was higher in ipsilateral gastrocnemius muscle than in that in contralateral. Results are presented as fold change. ***: *p* < 0.001 vs. contra. Fold change values were calculated as 2^ΔΔCt^.

**Figure 8 ijms-24-03765-f008:**
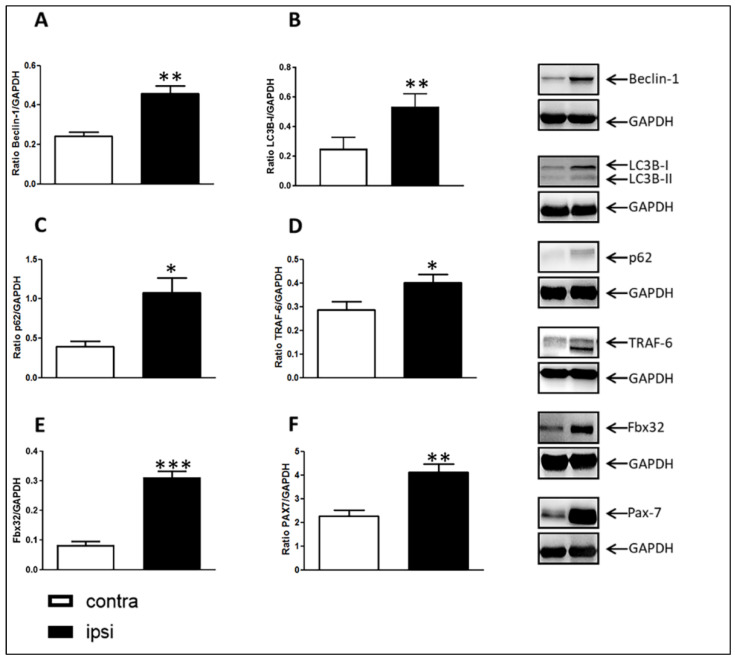
Western blotting images and densitometric analysis of markers analyzed in gastrocnemius muscle on day 28. Protein expression of Beclin-1 (**A**), LC3B-I (**B**), p62 (**C**), TRAF-6 (**D**) and Fbx32 (**E**) markers were higher in injured ipsilateral GC compared to those of contralateral muscle. Densitometric analysis revealed that Pax-7 expression levels (**F**) were also higher than those in contralateral muscle. Results are expressed as mean ± S.E.M. *: *p* < 0.05, **: *p* < 0.01, ***: *p* < 0.001 vs. contralateral.

**Figure 9 ijms-24-03765-f009:**
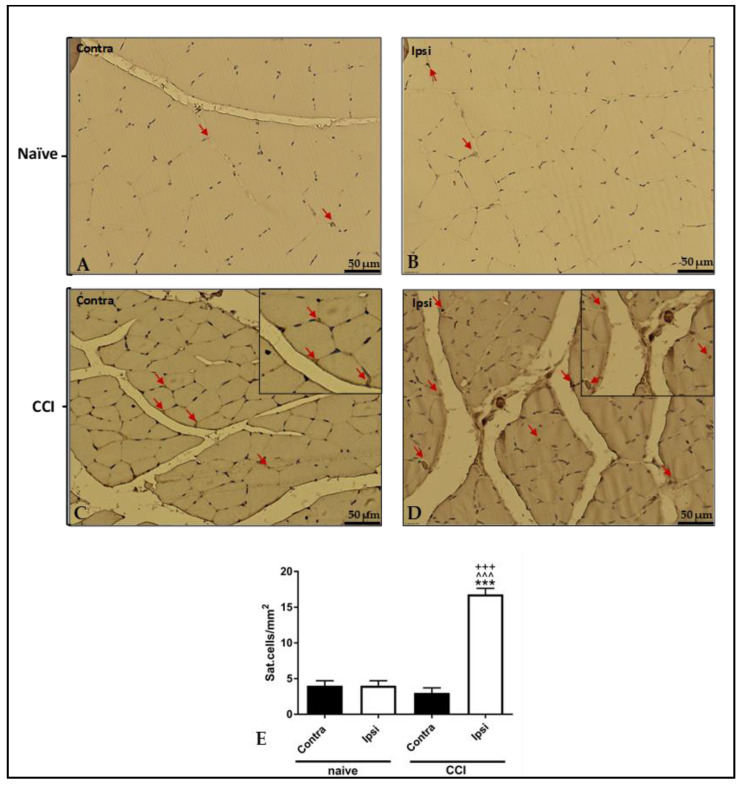
Results of immunohistochemical staining. Representative images of Pax-7 immunostaining in the ipsilateral CCI GC (**D**) show a higher number of Pax-7 positive satellite cells than that observed in the contralateral muscle (**C**) and in the contralateral (**A**) and ipsilateral (**B**) GC of naïve rats, as shown also in graph (**E**). Arrow = satellite cells. Analysis of immunohistochemical staining intensity in contralateral and ipsilateral gastrocnemius. Sections tissue was independently counted by three researchers and was averaged and depicted. The results are expressed as mean ± S.E.M. ***: *p* < 0.001 vs. contra CCI; ^^^: *p* < 0.001 vs. contralateral naϊve; +++: *p* < 0.001 vs. ipsilateral naϊve.

**Figure 10 ijms-24-03765-f010:**
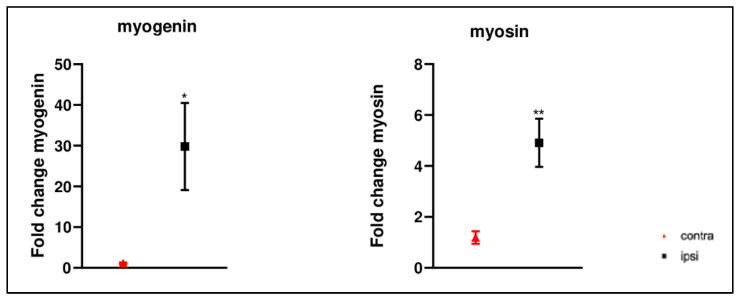
Results of real-time PCR. *Myogenin* and *Myosin* gene expression was higher in ipsilateral gastrocnemius muscle after chronic constriction injury compared to that of the contralateral. Results are presented as fold change. *: *p* < 0.05 vs. contra; **: *p* < 0.01 vs. contra. Fold change values were calculated as 2^ΔΔCt^.

**Figure 11 ijms-24-03765-f011:**
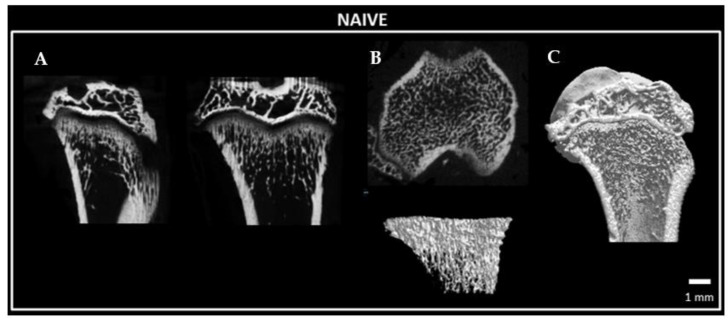
Representative micro-CT images showing trabecular architecture of the tibia bone of a naïve rat. (**A**) Coronal, sagittal and trans-axial sections; (**B**) Three-dimensional trabecular architecture; (**C**) Three-dimensional architecture of the distal tibia bone in coronal view. Scale bar: 1 mm.

**Figure 12 ijms-24-03765-f012:**
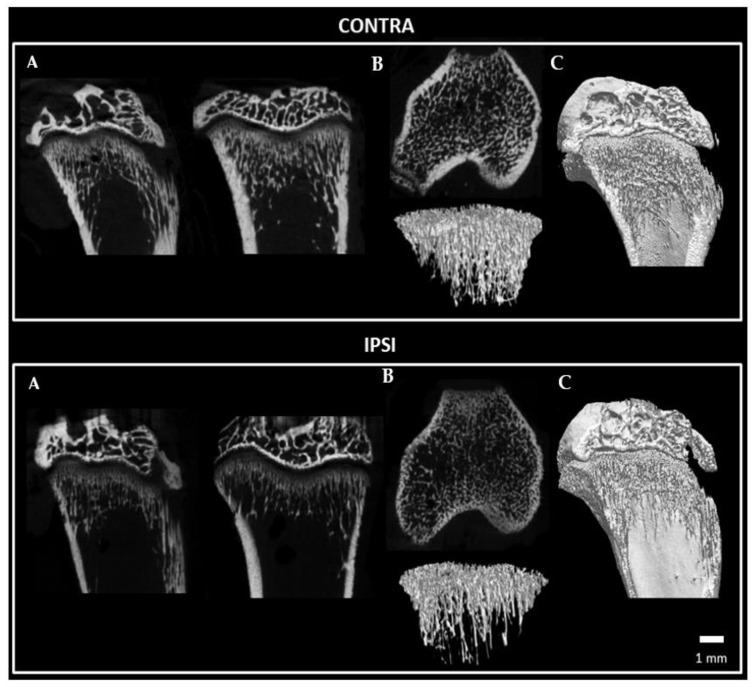
Representative micro-CT images showing the modified trabecular architecture of the tibia bone in a CCI rat. (**A**) Coronal, sagittal and trans-axial sections; (**B**) Three-dimensional trabecular architecture; (**C**) Three-dimensional architecture of the tibia bone in coronal view. Contralateral (**Top**), Ipsilateral (**Bottom**). Scale bar: 1 mm.

**Figure 13 ijms-24-03765-f013:**
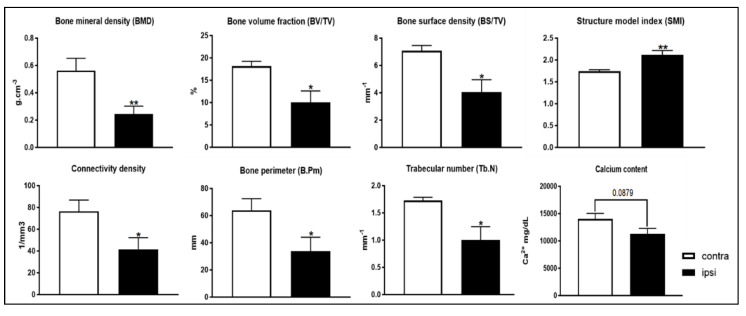
Bone structural parameters and calcium 4 weeks after surgery. Analyzed bone parameters were lower in ipsilateral tibia of CCI rats; the structure model index was higher; and the bone calcium content showed a decreasing trend. The results are expressed as mean ± S.E.M. *: *p* < 0.05, **: *p* < 0.01 vs. contralateral.

**Figure 14 ijms-24-03765-f014:**
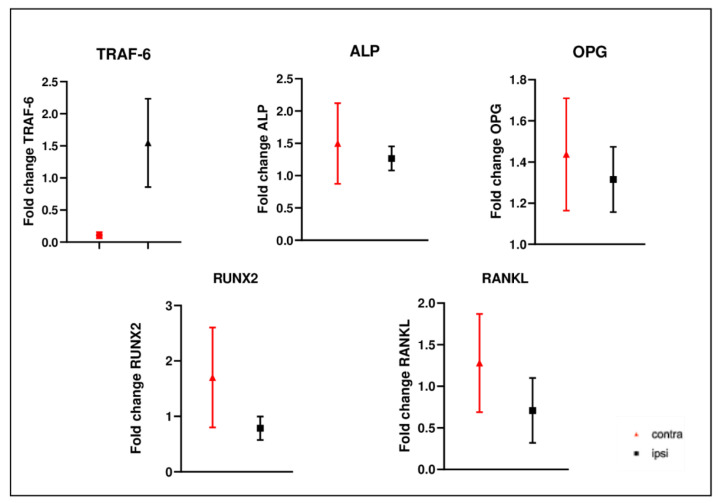
Results of real-time quantitative PCR in femurs. *TRAF-6* gene expression was up-regulated in ipsilateral bone; instead, *ALP*, *OPG*, *RUNX2* and *RANKL* gene expression were reduced. The results are presented as fold change. Fold change values were calculated as 2^ΔΔCt^.

**Figure 15 ijms-24-03765-f015:**
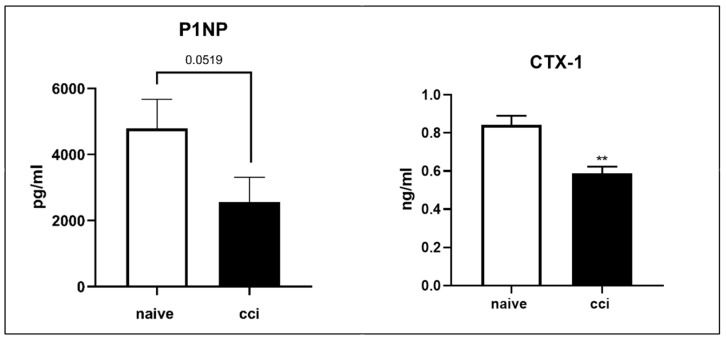
Serum markers P1NP and CTX-1 levels 4 weeks after surgery. CTX-1 serum levels were lower in CCI rats compared to those of naïve animals. P1NP showed a trend toward reduction in rats that underwent surgery. The results are expressed as mean ± S.E.M. **: *p* < 0.01 vs. naïve.

**Table 1 ijms-24-03765-t001:** Primers of genes used for real-time PCR on femur bones.

Gene	Forward Primer	Reverse Primer	Ref. Gene
*RANKL*	ACTTTCGAGCGCAGATGGAT	GCCTGAAGCAAATGTTGGCG	NM_057149.1
*OPG*	CCCAACGTTCAACAACCCAA	GGGCGCATAGTCAGTAGACA	NM_012870.2
*RUNX2*	CACAAGTGCGGTGCAAACTT	TGAAACTCTTGCCTCGTCCG	NM_001278483.1
*ALP*	TTGCTAGTGGAAGGAGGCAG	CATTGTGGGCTCTTGTGGGA	NM_013059.2
*TRAF-6*	ACTTGATCTCGGAGTGGTGC	CGTGACAGCCAAACACACTG	NM_001107754.2
*Β-ACTIN*	CCGCGAGTACAACCTTCTTG	CGTCATCCATGGCGAACTGG	NM_031144.3

## Data Availability

All data supporting the findings of this study are available within the paper and within its Appendix A published online.

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
