# Peer review of "Pathophysiological Aspects of Muscle Atrophy and Osteopenia Induced by Chronic Constriction Injury (CCI) of the Sciatic Nerve in Rats"

_ijms, 2023, doi:10.3390/ijms24043765_

Round 1
Reviewer 1 Report
I have read carefully the manuscript entitled “Pathophysiological Aspects of Muscle Atrophy and Osteopenia Induced by Chronic Constriction Injury (CCI) of The Sciatic Nerve In Rat” submitted to IJMS as an original article. Some points should be addressed before the manuscript can be considered for publication.
The main concerns are as follow:
A. There are many discrepancies in the nomenclature used throughout the manuscript. For example, just in fig. 7: Autophagic marker LC3 vs LC3B-I or Ubiquitin-protein ligase vs Fbx32. In manuscript, the first is also referred as LC3B-I, the latter as as Atrogin 1 (or Atrogin-1 or antrogin-1). Pax-7 or Pax7? Traf-6 or traf6 ? It must be unified.
B. Please pay attention to whether you are writing about genes (and their expression) or proteins. Sometimes it is not easy to follow. Genes should be in italics.
C. The results section needs significant improvements:
1. Start presentation of the results with body weight/composition data, not biochemistry.
2. Move 2.5 section before 2.4, as we first have transcription (mRNA fold change), followed by the translation (protein level).
3. All figure captions must be improved, as all figures with their corresponding captions must be self-explanatory. All abbreviations used in figures should be explained in figure caption.
4. In submitted manuscript, weights are presented in fig 2 not fig 1.
5. Some data in Fig 4 is missing. verify representative H&E images.
6. Why different presentation of mRNA fold change ? in fig 8, fig 10 and fig. 13 (please see y-axis). If fig 8: relative mean level - dCt, relative to what? If to contra, as in Fig 10, It should show no fold change for contra data (fold change 1). Similarly figure 13 caption does not give any answer (relative to what?).
D. Please attach in supplementary file uncropped WB images, as, according the information in instructions for authors "original, uncropped and unadjusted images should be uploaded as Supporting Information files ".
Minor comments:
L33 UPS – all abbreviations used in abstract should be explained
L48 .. and many other chronic diseases, including chronic inflammation of liver or pancreas, immobilization, prolonged bed rest etc.
L64 “well-investigated model” - reference(s) needed
L88 please add the information that it is a rat-based model study
L130-131 correct decimal separator to dot.
L347 “type II MHC” can be confused with MHC Class II (Major Histocompatability Complexes class II), sometimes written as “type II MHC”. Consider changing to myosin heavy chain type I and II or MHC I and MHC II, as in [73]
L367 “serum free (ionized) calcium”
L400 correct to “substance P”
L412 correct “precursor, procollagen, “ to “ precursor, type I procollagen”
L437 “bone blood inflow”
L457 There you state that rats were mature, in discussion in L224 that they were young. Please provide the information about rats age (months).
L520 “1% Triton X-100”
L521 chemical formula – subscript
L644 trends to statistical significance should be defined
Author Response
We thank the reviewer for the comments. We send the file with the answers.

Reviewer 2 Report
The authors of this manuscript aimed to evaluate if Chronic Constriction Injury (CCI) of the sciatic nerve in rat can be a valid model to study muscle atrophy and consequent osteoporosis. The introduction of the manuscript is relevant and gives sufficient information about the previous studies findings which leads to the current study rationale. To accomplish the aim of this study, the authors used many techniques (body weight, body composition, MRI, CT, Histological, immunohistochemical analyses, as well as qRT-PCR, and western blot). The methodology is generally appropriate, well-presented and organized in a logical way, beginning with the choice of a model, and continuing on through the selection of procedures, stains, analytical tools, and statistical methods. They concluded their work by recommending that the sciatic nerve constriction could be a valid approach to study muscle-bone crosstalk and to identify new strategies to prevent osteosarcopenia.
The study is so interesting, and the manuscript is clear and well-written. However, the authors need to give more clarifications about the following comments:
General comments:
1. Kindly make sure if all acronyms and initialisms are spelled out followed by acronym/initialism in parentheses the first time they appear in the abstract, media items, and the body of the article. E.g. line 33 UPS.
2. Result section: for mostly all the sections of the results, you need to introduce (one or two sentence) the markers or protein you are assessing. Why you decided to assess those proteins (e.g. in 2.1 section (line 92), you just mention if these protein CK, GOT, GPT increased or decreased and the reader has to go to the discussion to understand why you decided to assess these proteins).
3. Try to use the same name of the protein all over the manuscript. E.g GOT, GPT are used these names in the results and then most of the time in the manuscript you used AST and ALT. Same for Atrogin (Fbx32).
4. For gastrocnemius muscle, please either use GC or the full name in all the manuscript.
5. Please justify why you decided to use 58 rat in your study, do you use a calculation formula to reach to this number or based on a previous reference.
Minor comments:
1. Line 92: how many replicates (animals) were considered to reach the results. Please write the full names of the CK, GOT, GPT
2. Line 98: Make sure that you refer correctly to the pictures, please review.
3. Line 118: cannot find the figure that you are describing (Figure 4G demonstrates something different).
4. Line 124: Figure 4, This figure needs careful revision and review how you cite it in the results.
5. Line 127-128: this description has no pictures in figure 4, please review.
6. Line 130: the three up arrows are not used in the figure.
7. Line 134: You mean Figure 6, please review.
8. Line 136: Figure 5: It will be a good addition if you explained exactly what the readers should see in these MRIs, you should put arrows to indicate the structures.
9. Line 174: Please review the legend (no need to add (figure 9 in the figure description).
10. Figure 9: You should label different parts of the figures with A, B,....
11. Figure 4 and figure 9: please make sure to increase the scale bar in the histological and immunohistochemical pictures as it is not visible.
12. Line 187: Based on these CTs, can we conclude that those rats became osteoporotic. if yes, please mention this clearly in the results.
13. Line 188-190: How you score or calculate these parameters?
14. Did you do the CT for bones from all rats, if not, your statistical analysis based on how many CT scans?
15. Figure 12: make sure you put the title of the axes of all figures (structure model index figure).
16. Line 254, 255, please review.
17. Line 368: Parathyroid hormone NOT parotid hormone, Please review.
18. The discussion is very strong, the authors did an excellent job, however, you should add the limitations of the study at the end.
19. Line 486: how many samples did you collect (from each rat ??)? Please mention
20. Line 496: body fluids: You should include this also in your results especially that you already have the data.
21. Line 518: How many replicates you did for each sample?
22. Line 545: How did you calculate the cross-sectional area? which program did you use. Moreover, this analysis is based on how many slides (400 myofibers in a field? in a slide? per rat??) please clarify.
23. Line 547: in immunohistochemistry: What about the negative control? How you make sure your staining is due to the primary antibody used?
24. Line 636: please make this part as a separate section
Author Response

(The authors gave the same response as above.)

Round 2
Reviewer 1 Report
Thanks to the Authors for their diligent attention to the comments made